# Adolescents and Trust in Online Social Interactions: A Qualitative Exploratory Study

**DOI:** 10.3390/children10081408

**Published:** 2023-08-18

**Authors:** Elisa Colì, Marinella Paciello, Ernestina Lamponi, Rubina Calella, Rino Falcone

**Affiliations:** 1Institute of Cognitive Science and Technologies, Italian National Research Council, 00185 Rome, Italy; elisa.coli@istc.cnr.it (E.C.); rino.falcone@istc.cnr.it (R.F.); 2Faculty of Psychology, Uninettuno Telematic International University, 00186 Rome, Italy; ernestina.lamponi@uninettunouniversity.net (E.L.); rubinacalella74@hotmail.com (R.C.)

**Keywords:** online trust, adolescents, qualitative method, in-depth interview, deductive-inductive content analysis, NVivo

## Abstract

Social media have become increasingly embedded in adolescents’ daily lives. Although these contexts have been widely studied, how trust in online relationships is built among adolescents is still an unexplored issue. By adopting the theoretical socio-cognitive model of trust, this study aims to explore the components of online trust as far as today’s teenagers are concerned. The study involved 10 adolescents aged between 12 and 18 (M = 15.5). The data were collected using individual semi-structured, audio-recorded, and faithfully transcribed interviews. A deductive-inductive content analysis carried out with the NVivo10 software was performed on the textual material. Results show that adolescents seem to be aware of online trust value in “selecting” peers to be trusted. To protect themselves from the risks they are exposed to, they choose to interact with peers/friends who are already known in real life or are similar to them in terms of interests, ways of thinking, passions, and age. Additionally, others’ competencies and willingness play an important role in adolescents’ evaluations and decisions to rely on others online. The results of this study could be useful for developing awareness-raising interventions on the risks that adolescents are exposed to in order to promote “safe” relationships of trust and emphasize the possible positive use of technologies (e.g., by building online trust relationships using peer “safe” models).

## 1. Introduction

Trust is a key element in the formation and maintenance of interpersonal relationships and human interaction [1,2]. It reflects the outcome of internal mental processes [3] encompassing both cognitive and social aspects such as representations of the other involved in the interaction in terms of competence, dependence [4] as well as common membership in and affiliation to a salient social group [5].

During adolescence, trust has been attested as an important dimension related to the management of adolescents’ online opportunities and risks [6,7,8,9]. However, despite the fact that its role in understanding online dynamics is already recognized, how trust in online relationships is built among adolescents is still an unexplored issue. Understanding the antecedents of online interpersonal trust is important in light of the relationships of the new digital generations, as these relationships are often born and/or developed in online contexts. Social media, such as Instagram, TikTok, and Snapchat, have greatly transformed the structures and relationships of adolescents [10,11] and have become increasingly embedded in adolescents’ daily lives [12,13]. Although the literature on social media and adolescence is extensive [14,15], in light of our current knowledge, there is a complete lack of research on the social and cognitive elements that lead adolescents to trust others online. To fill this gap, this study aims to examine adolescents’ opinions based on the reasons why they trust others in online interaction contexts. We used the socio-cognitive theory of trust proposed by Falcone and Castelfranchi [3,16], which provides a model for understanding how adolescents express themselves about the determinants of trust through which they build interpersonal relationships in online contexts. This model could provide a useful framework for exploring online trust by identifying the social and cognitive components that can be recognized in the expressions used by young people when discussing what leads them to trust (or not to trust) others in online interpersonal interactions. A deeper understanding of these factors that contribute to trust and mistrust in online relationships among adolescents could help to bridge the gap in our understanding of how online trust is built and how much adolescents are aware of the knowledge and skills needed to safely navigate the complex world of online relationships. In the following paragraphs, after showcasing the extensiveness of literature on social media and adolescents, we provide a review of the literature on online trust during adolescence and the theoretical model of trust that we have used in this study to understand how adolescents come to trust others during online interactions.

### 1.1. Adolescents and Social Media Use: Reasons and Behavior

Nowadays, social media are the actual social labs of the new digital generations. The literature on adolescents and social media is extensive. The studies published in recent years mainly concern adolescents’ use, particularly reasons and behaviors in interactions on social media.

Concerning reasons, it is well-documented that young people are generally connected to satisfy their typical needs, such as the desire to interact with peers. Online adolescents share their thoughts and moods with close friends [17], and at the same time, social media facilitates the emergence of new relationships [18]. Another reason for adolescents’ social media use is the construction of their personal identity [19]. Indeed, social media allow adolescents to experiment with their own identities, identifying with a group or friend based on specific characteristics or common interests [20]. Moreover, social media provide the opportunity to construct an image for oneself by publishing and selecting certain information [21].

Regarding interaction modalities, while a part of the literature has focused on anti-normative behaviors, another part has focused on the constructive use of social media. The first group of studies showed that the time adolescents spend on social media and the problematic use can encourage online aggressive behaviors such as cyberbullying, hate speech, and online racism [22,23]. Cyberbullying can, in turn, have negative well-being outcomes for victims and perpetrators, including damaged relationships and heightened psychological distress [24]. Furthermore, experiencing cyberbullying victimization is a significant risk factor for suicidal behavior [25]. The second group of studies showed that constructive use of social media, for example, by adopting prosocial or supportive behaviors, promotes social inclusion and adolescents’ well-being [15]. Moreover, the active use of social media affects perceived social support, which in turn has a positive influence on the depressive moods of adolescents [26].

### 1.2. Online Interpersonal Trust and Adolescents

Trust plays a central role in face-to-face interactions as well as online ones. In particular, trust was identified as an element that can guarantee the success of these interactions [27], as well as the precursor of active engagement in online environments [28]. As regards the definition of online trust, Beldad, De Jong, and Steehouder [29] pointed out that the classic trust definitions can be applied to trust in online relationships as well. Therefore, online trust and face-to-face trust would seem to be based on the same fundamental components.

As underlined by Koranteng et al. [30], although offline trust definitions are applicable in online settings, the situational elements that influence the formation of trust differ. For instance, exchange appears to be a common factor in both settings, but exchanges in offline environments are different from online settings [11]. Issues regarding physical distance are also different in these environments, and in particular, social media eliminates the geographical boundaries [10]. In addition, human network attributes such as non-verbal language on which trust is built in traditional environments are absent on social media. Thus, the lack of elements relating to face-to-face relationships reduces the richness of communication among members [31]. 

Regarding online trust in adolescence, the most recent studies, conducted with quantitative methodologies, have focused on the relationship between trust and certain behaviors that take place in virtual environments [32]. In particular, online interpersonal trust seems to play an important role in understanding adolescents’ use of social media, including problematic use, for example, smartphone use [8] and social media addiction [33].

Concerning the studies conducted with qualitative methodologies, while those involving the adult population are quite numerous [34,35,36], those focused on online trust in adolescence seem to be almost absent in the last five years. One of these is Gibson and Trnka’s study [37], which explored young people’s use of social media to give and receive support in informal peer networks and underlined the importance that young people give to trusting relationships as a prerequisite for engagement with online support.

With reference to the specific context of social media, the most recent studies were conducted on adults and focused on the relationship between specific forms of trust—that is, social trust, particularized trust, and institutional trust—and participation in online networks [38]. Others were aimed at understanding how social ties are linked to an economic measure of trust [39]. An interesting study is that of Koranteng et al. [30], which, using a sample of university students, proposed a model investigating the factors that promote trust among social media users. Results suggest that the Norm of Reciprocity, Social Interaction Ties, and Identification are significant factors that encourage trust among social media users. However, this is a quantitative study that, while identifying the determinants of trust, does not investigate the meaning they assume for the participants in the study itself.

As shown in this literature review, despite the increasing dissemination of studies on online trust, how trust is attributed in online environments, with a particular focus on social media and adolescence, is still unknown. This paper, therefore, seeks to expand the current literature on adolescents’ relationships with social media and trust, introducing a qualitative analysis methodology that appears to be absent in the study of online interpersonal trust in young people. It adopts the socio-cognitive model of trust, already successfully applied in order to measure different kinds of trust to investigate the factors that promote the formation of trust among adolescents interacting in social media. Furthermore, in the complex online scenario, the understanding of online trust could provide a more comprehensive lens in order to capture the evolution of different social behaviors and interpersonal dynamics and understand how they can move together toward adaptive or maladaptive paths.

### 1.3. The Socio-Cognitive Model of Trust

The concept of trust is central to our study, and in particular, we intend to explore the elements that contribute to the establishment of trusting relationships in the context of online interactions between adolescents. To achieve this, we used the socio-cognitive model of trust as a guide [3,16]. According to this model, trust is a relational construct that involves a trustor (X), that is the one who must trust someone else to carry out a certain task, and a trustee (Y), that is the subject on which trust is posed (with respect to that, more or less specialized or generic, task). When a relationship of trust is established, the trustor entrusts the trustee with that task, which necessarily will imply actions by the trustee in an attempt to obtain the expected result. The relationship of trust promoted by the trustor is aimed at achieving a goal, namely the one that led to the definition of the relationship of trust itself. The trustee’s actions will take place in a given context, an environment that will certainly influence the actions themselves.

In particular, trust is based on two fundamental mental ingredients, namely goals (that is, the mental representation that identifies the desired state) and beliefs, the most important of which are competence and willingness attributed to the trustee. When the trustor delegates a task to the trustee, he/she must believe that the same is capable of doing things that he/she needs and that he/she has skills (competence belief). The trustor must also believe that the trustee intends to perform the task and that he/she will perform it (willingness belief). Therefore, the first belief refers to the possession of skills, competencies, and support tools to reach a goal, while the second refers to the possession of the attitudes of activation towards the task, which is intentionality, disposition, and motivation. Adding to these, there are other relevant beliefs, including the unharmfulness belief, which refers to the possibility that, voluntarily or involuntarily, the subject may damage the achievement of the goal; the dependence belief, which refers to the possibility of being able to reach the goal more or less autonomously; and on the trustor’s other experiences belief of trust in the context, relative to how much the context is able to positively or negatively interfere with the achievement of the goal. Any action of trust implies a bet and, therefore, a risk. The trustor, in fact, with his/her act of trust, makes him/herself dependent on the trustee and in some way exposes him/herself, becoming vulnerable to the trustee’s actions. In fact, there is a risk that the goal is not achieved, resulting in its failure and generating waste of material and not material resources.

The socio-cognitive model considers trust as an intrinsically dynamic phenomenon, which changes over time based on the changes that occur in the sub-components on which trust is based. In particular, it is a dynamic phenomenon depending on the trustor’s previous experiences of trust (with the same or with other agents, on specific or not specific tasks, in some environments rather than in other environments). Trust also changes when the other sources on which it is based change (for example, people’s way of thinking or the reputation of the trustee with whom they interact).

This model has the advantage of having identified a conceptual core capable of taking into account the most relevant processes that may lead to trust in the other person. Furthermore, it has operationalized the construct of trust by identifying the components on which it is based and the process, which, starting from an evaluation and going through a decision, leads to the actual act of trust. Initially applied in agent simulations [40], it has recently found wide application in the social sciences guiding the construction of scales aimed at studying trust in institutions [41] and in the doctor-patient relationship [42]. It has also been applied to the preliminary measure of adolescents’ online/offline interpersonal trust [43].

Precisely because of its characteristics, this model could lend itself very well to an in-depth study of trust, guiding the exploration of the ingredients that contribute to the attribution of trust in the context of adolescents’ online interactions. Knowing the point of view of young people concerning the beliefs theorized in the socio-cognitive model could prove particularly useful in order to fill the lack of studies in this area and to provide guidance to those who work with adolescents, for example, suggesting which elements should be considered in order to promote safe online behaviors.

### 1.4. Purpose of the Study

Starting from the considerations made above and related to the necessity to fill the gap in the literature on trust in adolescents’ online relationships, the study aims to explore the trust that adolescents place in others with whom they interact on social media. Specifically, we analyzed the socio-cognitive determinants of trust, with particular reference to beliefs (of competence, willingness, unharmfulness, dependence, and context) on which adolescents’ online trust is based. In particular, we tried to answer the following questions: From adolescents’ point of view, what skills should the others to whom they rely online have? On which basis should an adolescent evaluate someone online as available and as non-dangerous? What are the factors that make adolescents feel dependent on others online? Finally, what are the context-related factors that foster or hinder the relationship of trust? Therefore, the proposed study is exploratory. It does not have the purpose of verifying hypotheses, but rather, it aims to explore a still little-known phenomenon, namely that of trust in online relationships, starting from the point of view of young people who interact on social media.

## 2. Method

The researchers adopted a qualitative methodology [44,45], which seemed particularly suited to the aim of the study, which is to explore the elements on which trust in online relationships is based. From a theoretical point of view, the socio-cognitive model of trust informed the entire study, starting from the research questions and from the definition of the interview outline, followed by the data analysis up to the reading of the results. It should be noted that one of the founders of the model itself was a member of the research group, and this ensured the constant presence of an expert point of view on the theory of trust. We collected data using a semi-structured interview, which seemed to be the most appropriate methodology for the research objectives, having the very purpose of understanding the participants’ point of view [46], representing a flexible and powerful tool to capture the voices and the ways people make meaning of their experiences [47].

As regards the analysis of the collected data, we used qualitative content analysis, which aims to promote knowledge and understanding of the phenomenon under study [48]. In particular, on the basis of the purpose of our study, we chose to use a mixed deductive-inductive approach to content analysis [49]. Therefore, we first created a coding system based on the main determinants of trust identified using the socio-cognitive model. Then, we analyzed the text of the interviews within this coding system (deductive approach). Subsequently, guided using an inductive approach, researchers created a system of categories based on content similarity for each determinant of trust.

### 2.1. Participants

The study involved ten adolescents, five males and five females, aged between 12 and 17, with a mean age of 15.5 years. The majority of participants (six) had a middle school education level, and the remaining four had a lower school education level. Regarding the main social media used to interact, everyone used both Instagram and WhatsApp, while only one interviewee used Facebook and another one used Snapchat. Researchers recruited participants in a small town in the South of Italy using non-probabilistic cascade sampling. In particular, we started from the personal network of one of the researchers, including a teacher and the head of a parish center in the town, who have identified young people willing to take part in the research. Considering that the aim of the study was exploratory and that the researchers’ interest was to know the point of view of young people on online interpersonal trust, the only conditions for inclusion in the sample were the age and the use of one or more social media.

Researchers adequately informed the participants and their parents about the research, with particular reference to the study objectives, research tools, procedure, possible limitations, and benefits of the study, as well as data processing in accordance with current legislation. They were also informed about their rights, such as the possibility to withdraw from the research at any time without having to provide any explanation or the right to request access, rectification, or deletion of personal data. After reading the privacy policy and the informed consent, they freely expressed their willingness to participate in the study by signing both they and their parents the informed consent form. Therefore, participation in the research was voluntary, and no incentives were offered to the participants, as the research took place without access to any kind of funding.

For the conduct of this study, researchers received approval from the ethics committee of the University of one of the authors and the Faculty Council on 1 February 2018. Moreover, in order to ensure the privacy, safety, and confidentiality of the participants, several steps were taken. First of all, we identified the figures in charge of data processing authorized by the Institute researchers are connected with. Second, we processed the data in accordance with the privacy laws and in accordance with the Legislative Decree of 30 June regarding the protection of personal data and the European Privacy Regulation EU 2016/679 (GDPR) by guaranteeing the anonymity of the participants. We also guaranteed participants’ privacy by assigning a code to the subject, and we kept sensitive material on the PC of one of the researchers, protected using a password. Only the research group, specifically the researchers identified as data processors, had access to these data.

### 2.2. Interviews

We collected data using individual semi-structured interviews. In particular, researchers developed an ad hoc interview guide following the interview steps by Smith and collaborators [50] and paying particular attention to the formulation of the questions, as suggested by McNamara [51]. The definition of the questions had as a theoretical reference the socio-cognitive model of trust and was aimed at exploring the model’s beliefs and how they are applied in online relationships between adolescents.

In particular, the interview opened with a question aimed at identifying the social media used by adolescents to interact with others online, followed by an opening question on the trust-specific topic aimed at understanding how young people conceptualize online interpersonal trust and on which elements they base it. Subsequently, the specific components of trust, as theorized in the socio-cognitive model, were explored starting from the expectations placed on the other online—which are considered a prerequisite on which trust is based—and following with the competencies recognized in the other, his/her willingness, the perception of unharmfulness in the relationship, the dependence on the other and finally the context in which relationships take place. At the end of the interview, researchers asked participants to fill out a very short questionnaire containing questions regarding personal information (age, gender, educational qualification) and information regarding the main social media used to interact.

We conducted the interviews between December 2019 and January 2020 in a room made available by the parish center of the town. As suggested by Turner [52], we chose a comfortable environment where participants did not feel restricted or uncomfortable sharing information. As necessary, we asked spontaneous follow-up questions to clarify issues raised by the interviewee [53].

All the interviews, lasting an average of about 45 min, were audio-recorded using a high-sensitivity digital voice recorder, and subsequently, the research group of the article faithfully transcribed them. Researchers continued with data-gathering until a point of saturation was reached when no new ideas emerged. We kept the recordings and related transcripts and the file containing the personal information of the participants on the PC of one of the researchers, protected by a password.

### 2.3. Analysis

We analyzed the textual material deriving from the transcription of the interviews using deductive-inductive content analysis, following the procedure described by Elo and Kyngäs [54]. In particular, we chose a deductive approach [55] because our study was based on an existing theoretical model. Together with the deductive approach, we used the inductive one [56] because it is particularly suitable to describe adolescents’ in-depth perceptions concerning each of the theoretical trust model’s beliefs. Therefore, the elements captured in the study are partly emanated from the referring theoretical framework and partly from collected data.

After the interviews were made and transcribed, we imported them into NVivo [57], the software that guided the data analysis. Before proceeding with the analysis of the text, we read and reread the interviews several times. This step allowed us to start diving into the data [58,59] and identify ideas and concepts relevant to the research questions [53]. Another preliminary step concerned the definition of a unit of analysis, which, as suggested by Guthrie and collaborators [60], we identified in the unit of content/meaning.

Only at this point, the actual analysis process started. In the first step, in line with deductive content analysis, we created an ad hoc classification system starting from the determinants of the theoretical model of trust [61]. Then, we proceeded with the first coding of the interview texts. In particular, we highlighted and classified the different portions of the text into nodes, which had extensive labels that reflected the content of the node itself [62]. We then inserted the created nodes, called free nodes, in the classification system and, in particular, in the node competencies, willingness, unharmfulness, dependence, and context, which we will call macro-categories.

This initial deductive analysis made it possible to guarantee the correct positioning of the contents that emerged from the interviews in the respective beliefs of the model without assuming that everything that emerged in the investigation of an area of the interview necessarily referred to that area. For example, when we asked participants to talk about skills, some reported elements that, according to the theoretical model, referred to willingness, so we placed them in the “willingness” area. The first author initially developed this first coding work and subsequently shared it with the research group in numerous meetings aimed at comparing results. During these meetings, thanks also to the presence in the working group of the founder of the socio-cognitive model, the researchers repeatedly reviewed the classification of the text portions and their placement in the various macro-categories until arriving at a coding shared by all researchers [63,64]. This first phase, which required an extensive amount of time, allowed the whole working group to acquire the maximum possible confidence in the data and was also of fundamental importance in order to proceed to the following inductive coding work.

Subsequently, in line with the inductive approach, we opened and read the free nodes present in each macro-category, then we grouped them on the basis of the similarity of content and inserted them into higher-order categories created ad hoc, the so-called main categories [65,66]. For example, as far as competence macro-category is concerned, we inserted all the free nodes that referred to rules of communication and interaction into a node created ad hoc, which is the main category called “social norms”; we placed nodes that referred to competencies related to self-presentation in the main category called “identity” and so on.

Further grouping work was carried out on the nodes in each main category, which led to the identification of first and, when necessary, second-level subcategories [53]. For example, as regards the “competence” macro-category, within the main category of “social norms”, we identified the first level subcategory called “communication”, within which in turn we identified the second level subcategories, namely, ”content” and “methods”.

In this way, for each area explored in the interview, researchers generated a hierarchical, tree-like system of nodes, which made it possible to simplify the complexity of the emerged contents. This system was made of main categories or higher order nodes (parent-nodes) and more specific sub-nodes or subcategories (child-nodes) [67]. The use of the NVivo10 software ensured the accessibility, in each phase of the analysis process, of the complete text associated with each identified category, promoting transparency of the analysis process [62]. Among other things, the use of NVivo has proved particularly useful for its features, including being able to export the classification system to Excel. This last function has allowed researchers to easily discuss, each time, the text classification system. In particular, the accuracy of the analysis was guaranteed by a categorization work carried out individually and in parallel by the researchers, who subsequently confronted each other, questioning several times the emerged categories’ system until reaching a shared categorization [68]. The team of researchers, moreover, during the different phases of the analysis process, availed itself of the advice of the founder of the socio-cognitive model, discussing with him from time to time the placement of the categories in the different areas of the model.

## 3. Results

The results are organized in paragraphs. Each paragraph is referred to one of the beliefs, theorized using the socio-cognitive model, that contributes to the trust. In particular, using the qualitative content analysis, we have explored the adolescents’ points of view about each component of trust, referring to online relationships. In line with the research objectives, we have reported only the findings related to the beliefs of competence, willingness, unharmfulness, dependence, and context. The results are reported descriptively and integrated with the sentences spoken by the participants, identified using a pseudonym.

### 3.1. Competence

The adolescents interviewed point out different skills that promote trust in the others with whom they interact online.

Some, such as Ernestina, refer to skills related to the management of online communication:

You need to know how to calibrate well what is written and shared on the net and filter well what you want to say … even the way you write, you should write in full, without abbreviations, and in a clear way.

For Ernestina, therefore, the other with whom one interacts must know how to manage not only the content of the communication but also the form. Others, such as Rino, on the other hand, highlight aspects relating to interaction management:

I am going to see if he/she interacts often and constantly without offending anyone in the stories … In short, he/she should know how to be a calm person even in the comments of posts such as those for football teams. If he/she supports another team, for example, he/she should not create quarrels.

For Rino, on the other hand, the other must know how to manage the interaction not only from the point of view of quantity but also of quality. In particular, in addition to being frequent and constant, the exchange should take place with respect for the other and for points of view/passions different from one’s own.

Further skills, which emerge, for example, from Davide’s words, refer to the ability to use social media constructively:

He/she must be able to often share interesting topics that lead to discussions, such as youth topics, useful for school or related to the presentation of health products, such as diet products or make-up ones.

Davide underlines the importance of an “engaged” use of social media, which favors comparison and the acquisition of new knowledge, drawing attention to the information and training potential that social media can have for adolescents.

Alongside these, other skills emerge, albeit shared by a smaller number of adolescents, relating to the use of applications connected to social media and self-presentation. As regards this second aspect, young people focus on the ability to be original and to provide a good description of themselves, an element Marco talks about:

Maybe he/she should put a description of him/herself full of elements to give an idea of how he/she is, to make people understand what kind of person he/she is, the hobbies he/she has, a bit of a story of him/herself.

This excerpt emphasizes the competence relating to the definition of one’s online profile and the importance of providing as many elements as possible that can reassure the identity of the person with whom one interacts.

### 3.2. Willingness

Teens talk about a number of elements that are indicative of how much others are really motivated and willing to accomplish the adolescent’s purpose, namely to establish a positive online relationship.

Most of them refer to the presence of the other, which can be deduced from a set of elements highlighted by Davide’s words:

I pay attention if he/she replies immediately to messages, comments positively on my photos, responds to surveys every time, and always liking my photos … or if he/she contacts me to spend a different day, maybe more fun, like when I am down for some reason.

Davide underlines that the willingness of the other in terms of presence can be deduced from an interaction that occurs through comments and responses to photos, stories, and surveys, but also through simple likes. It also shows how the interaction should respect two characteristics, and in particular, it should be immediate and frequent. The others’ presence can also be deduced from real and concrete proximity, which manifests itself in times of need, moving from the virtual world to the real world.

Some adolescents instead focus on how communication exchanges should take place in the online environment. Rubina, for example, underlines how online relationships should be guided by kindness and mutual respect, while Marinella highlights the importance of respecting privacy, as it emerges from her words:

You understand if a person wants something positive from you if he/she does not ask you uncomfortable questions; maybe he/she just asks things to get to know you better but positively, without going into the most intimate sphere … I know that there are many guys who ask girls for intimate photos and in my opinion, this is already a way to understand that that person does not want anything positive from you.

Marinella, in particular, reports how remote interaction should take place through a series of questions that foster authentic knowledge of the other without putting him/her in difficulty by invading, for example, his/her intimate and private sphere. Finally, a smaller number of adolescents, such as Marco, refer to the trustworthiness of the person they interact with:

If they first say one thing and then another, it gives me the idea that what they say may be false.

From Marco’s words, it emerges how, in order to attribute trust in online relationships, it is necessary to pay particular attention to the content of the communicative exchanges, as their coherence can be indicative of the reliability of the other with whom one interacts.

### 3.3. Unharmfulness

As regards the factors which, by fostering security, contribute to promoting a relationship of trust, most adolescents focus on knowledge of the other.

In particular, they relate to people who they already know in real life, but also to a knowledge built up over time, as expressed by Davide:

I do not think you need to look at a profile with nice photos, but you need to get to know him/her slowly, interacting over the days and discussing via chat or personally…You should never be sure about someone you do not know.

From Davide’s words, it emerges that the process of attributing trust in online interactions seems to be guided by a certain caution. In particular, previous knowledge or knowledge built over time would seem to act as a reassuring element that goes beyond the information conveyed through the profile.

Another element that emerges from the interviews is related to a sort of social closeness, as highlighted by Rino:

When I trust him/her, it is because perhaps I have noticed that he/she is my age, that he/she has the same passions as me—such as football—and contacts in common, or my ways of thinking, for example, based on the comments he publishes in players’ posts.

From Rino’s words, it emerges how having elements in common, such as friendships, interests, and ways of thinking, but also a similar age represents a factor that fosters long-distance relationships of trust. Further elements that can be deduced from the profile analysis are relevant, as shown by adolescents during the interviews. Ernestina, for example, states:

If they are only online friends I do not know personally, I can see if they have a reliable profile and if they have many photos. If it is a well-kept profile with various personal photos, maybe they can be trusted, but partially, not completely. When I do not know them, I interact less; it’s a “half” trust. If they do not have personal photos, if they do not share posts if they have few followers, I imagine it’s a fake profile, and so I do not interact.

This excerpt highlights the fact that teenagers, interacting online with people they do not know in real life, act with caution and that the trust placed in them is always partial and well-considered. In particular, at first, they pay attention to profile analysis, looking, for example, at the quantity and type of shared photos, number of followers, and activity in terms of published posts. Later, the attention shifts to the interaction, in particular to the adopted communication method. Additional elements in this sense are introduced once again by Ernestina:

If I do not know him/her personally, I pay attention to the way he/she chats and writes, if he/she uses a language that I like or not if he/she is vulgar, or if maybe he/she wants to overstep the boundaries or if he/she’s intrusive.

Ernestina’s words highlight that adolescents, in evaluating the security of a relationship, focus on the way of writing, emphasizing the importance of using “clean” language, and on the way they interact, emphasizing the importance of respect for their times and boundaries.

### 3.4. Dependence

As to the issue of dependence on the other with whom one interacts online, some of the adolescents interviewed declared that they do not feel dependent. Most, however, depend on others essentially out of a need for visibility, as expressed by Rubina:

I depend on people who have more followers than me; in particular, I try to interact more with them, and I hope they are the ones who make more comments, reply to my stories, and like me. For example, I share a product and tag the company or person who advised me to buy it, and they, in turn, re-share my post, and someone notices my profile, and I have more requests.

From this excerpt, it emerges how adolescents are very practical/competent in the use of social media concerning the strategies implemented to gain visibility and expand their audience of followers, and how this need, however, leads them to depend on who has more followers than them.

Even the need for help and interaction makes the adolescent feel dependent on the other, as emerges from Elisa’s words:

Sometimes, if I do not feel with someone, I also feel a little empty, perhaps because I lack help to achieve some things.

From these words emerges the need to always be in contact with the person with whom one interacts, a need probably dictated by the intrinsic characteristics of social media, always and everywhere available, with a consequent sense of social emptiness and loneliness at the moment in which this contact should come to be missing or simply thinning out.

### 3.5. Context

The external context, that is, the virtual environment in which the relationship takes place, can foster or hinder the establishment of a relationship of trust.

Among the favoring factors, adolescents refer to relational and communicative accessibility and immediacy, as highlighted by Marinella:

It has positive aspects such as always keeping in touch, therefore also a more immediate relationship. If you have something to say, you expose it immediately; it is different in comparison with a long time ago when letters were used, and it took about a week.

From Marinella’s words, it emerges that one of the main characteristics of social media is to foster relationships since the internet connection is now always and everywhere accessible, allowing even distant people to always be in touch. Additionally, the importance of immediacy is underlined, as it allows contact and exchange in real time without waiting for a long time.

However, most adolescents focus on the hindering factors, among which they mainly highlight the difficulty in interpreting the real communicative intentions of the other, as underlined by Elisa:

Perhaps he/she posts a happy face, and, in reality, he/she does not even want to talk … I do not understand the real intention.

Elisa’s words seem to indicate that the interaction mediated by the internet and the consequent lack of face-to-face communication can cause misunderstandings and interpretative difficulties also on an emotional level. Fewer adolescents speak about anti-normative behaviors, which are related to the unauthorized dissemination of information, such as personal photos or confidential conversations, and identity theft, as described by Marco:

About a week ago, my Instagram profile was hacked; I had to delete my profile because there were my private things … I got an idea who could have done this … Before, I trusted this person, now I put more attention in relationships on the internet.

From Marco’s words, the dimension of risk closely connected to the use of social media and online interactions emerges again. This element encourages the use of caution in the management of remote relationships of trust.

## 4. Discussion

The exploration of the adolescents’ trust beliefs, as they have been theorized in the socio-cognitive model of trust, indicated that the competencies related to the rules of conduct to be kept in the online environment seem to prevail in the adolescents’ evaluations and decision-making to rely on online others. In particular, in the online context, the management of the communication itself becomes essential to avoid possible misunderstandings that on social media could be more frequent than in real life since the interpretative and emotional clues of the communicative content are certainly less numerous [30].

In line with the literature [69], the capability to manage online interaction in a respectful way also emerged as another fundamental element in the understanding component of trust in a virtual environment. Indeed, the interview analysis suggests that non-normative behaviors could be more difficult to manage in online interaction than in offline ones. For example, there is persistence over time of the negative information shared online or the possibility of quickly reaching a large number of unknown people.

Therefore, social media, as any tool, should be used in a respectful and constructive way; for example, they can promote and reinforce pre-existing healthy interpersonal relationships [8]. As shown by other authors [70,71], our data also suggest that adolescents’ social skills play an important role in setting responsible and healthy interactions, even in technologically mediated communication contexts. With regard to the establishment of new peer relationships, it is notable that among the skills reported by adolescents, those relating to self-presentation also emerge [72]; in particular, adolescents tend to trust those who present themselves in an original way by distinguishing themselves from the group and those who take care of their own profile by providing a self-accurate description. This finding can provide a new contribution to the existing literature focused on online self-presentation. Indeed, differently from the studies that underlined the adolescent’s tendency to conform to online models [73], it seems that when young people have to select people they can trust, they prefer to follow those peers able to present themselves authentically, accurately and originally.

In the domain of others’ willingness, from the adolescents’ point of view, the presence of the other is particularly relevant, consistent with proper forms of the virtual environment. In line with the literature [74], the online presence can be deduced from comments and replies to one’s posts/messages but also from simple likes [75]. Moreover, the interaction with the other should be immediate since; for adolescents, the virtual environment seems to require others to be constantly connected and prone to be immediately responsive. Despite that, the relationship also requires elements that make it more similar to that of real life [76]. For example, others’ availability in terms of support and stability or the possible quick transition from online to offline context represents the best place for providing greater intimacy and confidentiality.

Adolescents seem to be also aware that giving trust always exposes them to risks, especially in the virtual environment. In accord with previous studies [20], among the elements that adolescents indicate as priorities to protect their security, there is the choice to interact with peers/friends that are already known in real life or with more similar peers in terms of interests, ways of thinking, passions, and age. To this end, from the adolescents’ point of view, a series of elements can be extrapolated from a sort of profile analysis related to the other’s online self-image [21].

The characteristics of the interaction that takes place in the online environment represent the main factors that can favor or, at the same time, hinder trust relationships. If, on the one hand, the online context, due to its technological affordances, can guarantee accessibility and immediacy of relationships, allowing, for example, the maintenance of relationships at a distance, on the other hand, the lack of face-to-face cues can make understanding the others’ real intentions and emotions difficult. Additionally, this result is in line with previous studies [21,26].

## 5. Conclusions

Overall, the present study provides a detailed photograph of how trust in online relationships between adolescents can be configured, guiding the comprehension of the ingredients that contribute to the attribution of trust in online environments. In this sense, this study represents an attempt to fill the lack of qualitative studies on online interpersonal trust in young people. The usefulness of this study rests on its theoretical basis as well as on its application aspects. At the theoretical level, the present study offers a detailed picture of how trust in online relationships between adolescents is configured. It sheds light on the social and cognitive factors that contribute to the attribution of trust in virtual environments, enhancing the comprehension of this complex phenomenon. On the one hand, the study contributed to the advancement of the theoretical reference model, showing its potential application in online trust attribution processes. On the other hand, the study contributed to filling the literature gap not only concerning the knowledge of factors on which adolescents base their trust in social media but also concerning what these factors mean for young people. From an applicative point of view, knowing these factors can be useful for supporting young people’s relationships on social media. Specifically, the results of the present research could be used for developing awareness-raising interventions on the risks that adolescents are exposed to when they trust others in an online context in order to promote “safe” relationships of trust. For example, starting from factors that led adolescents to trust others online, it is possible to identify potential adolescents as preferred recipients of peer trust for implementing peer education interventions. These trusty adolescents could become active figures in promoting citizenship and responsible social media usage educative programs. In this sense, this study could emphasize the positive use of technologies if the building of online trust relationships between adolescents occurs through peer “safe” models of online interpersonal relationships [6,77].

Despite the potentialities, this study presents a number of limits. First of all, one relates to the sample, which involved young people with an age included in a rather wide range and which is geographically limited. The adolescents interviewed came from a small southern town, thus excluding young people who live in northern or central Italy and in metropolitan areas. Another limit is linked to the participation on a voluntary basis and to the methods used to contact young people, elements that may have added further bias. For example, some of the young people were recruited at a parish center, and this may have provided limited insight into the type of online interactions of a specific population. A further limitation is represented by the need to extend the number of participants. Additionally, the study reported here was developed following a specific theoretical model of trust, which guided the formulation of questions posed during the interviews. Consequently, the study necessarily focused on specific determinants of trust in online interactions.

Further research should be conducted by extending the sample from the point of view of geographical origin, keeping the age groups involved under greater control, and making use of more varied recruitment criteria to ensure the generalizability of the results. Furthermore, future qualitative studies could be developed to investigate if and how interactions on social media by adolescents have changed, with particular reference to the role of trust, following the COVID-19 pandemic, which has inevitably led to a series of changes in human relationship management, including the online ones [78,79].

## Data Availability

The dataset supporting the results of this study is available upon motivated request to the corresponding author [blinded]. The data are not publicly available because, although completely anonymous, their public disclosure is not specifically mentioned in the informed consent provided by the participants on the use of confidential data.

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
