# Peer review of "Adolescents and Trust in Online Social Interactions: A Qualitative Exploratory Study"

_children, 2023, doi:10.3390/children10081408_

Round 1
Reviewer 1 Report
Dear authors
congratulation to your paper.
Adolescents and Trust in Online Social Interactions: A Qualitative Exploratory Study.
The paper aims to explore the components of online trust for teenagers.
plus: very precise introduction (good example for other papers)
methods - analyses - : excellent job, very good analyses
I like: your vision of "further research" -
but It would be good to add: (the last part) CONCLUSION
---
It is important to cite similar topic papers to deepen the research :
Tkacová, H.; Králik, R.; Tvrdoň, M.; Jenisová, Z.; Martin, J.G. Credibility and Involvement of Social Media in Education—Recommendations for Mitigating the Negative Effects of the Pandemic among High School Students. Int. J. Environ. Res. Public Health 2022, 19, 2767. https://doi.org/10.3390/ijerph19052767
Maturkanič, P.; Čergeťová, I.T.; Králik, R.; Hlad, Ľ.; Roubalová, M.; Martin, J.G.; Judák, V.; Akimjak, A.; Petrikovičová, L. The Phenomenon of Social and Pastoral Service in Eastern Slovakia and Northwestern Czech Republic during the COVID-19 Pandemic: Comparison of Two Selected Units of Former Czechoslovakia in the Context of the Perspective of Positive Solutions. Int. J. Environ. Res. Public Health 2022, 19, 2480. https://doi.org/10.3390/ijerph19042480
Maturkanič, P.; Tomanová Čergeťová, I.; Konečná, I.; Thurzo, V.; Akimjak, A.; Hlad, Ľ.; Zimny, J.; Roubalová, M.; Kurilenko, V.; Toman, M.; et al. Well-Being in the Context of COVID-19 and Quality of Life in Czechia. Int. J. Environ. Res. Public Health 2022, 19, 7164. https://doi.org/10.3390/ijerph19127164
I recommend publishing your paper.
Author Response
Thank you very much for the positive feedback and the recommended papers that have been included in the current version of the manuscript.
Reviewer 2 Report
1. Some grammatical errors are observed in the manuscript.
2. Authors assert that "although virtual contexts and online dynamics are widely studied through different approaches, what is missing is an in-depth knowledge of elements on which adolescents ' trust relationships on social media are based". Authors should indicate how they arrived at this assertion through a convincing literature walk-through.
3. What is the relevance of the study? Authors should indicate the specific contribution of the study findings to theory and practice.
The quality of English Language is on the average. Moderate revision is required.
Author Response
We thank the reviewer for the prompt and improving feedback. We have revised the manuscript according to the provided suggestions, as indicated. We hope that the manuscript has now been improved
Please find below the details of our making changes:
- Some grammatical errors are observed in the manuscript.
We had the manuscript reviewed by an expert to improve the English
- Authors assert that "although virtual contexts and online dynamics are widely studied through different approaches, what is missing is an in-depth knowledge of elements on which adolescents ' trust relationships on social media are based". Authors should indicate how they arrived at this assertion through a convincing literature walk-through.
The sentence has been rephrased in the hope of making the concept clearer, and some references have been added.
Although the literature on social media and adolescence is extensive (Bozzola et al. 2022; Valkenburg et al. 2021), in the light of our current knowledge, there appears to be a lack of research on the specific elements that lead adolescents to trust others online
Es. Bozzola, E., Spina, G., Agostiniani, R., Barni, S., Russo, R., Scarpato, E., ... & Staiano, A. (2022). The use of social media in children and adolescents: Scoping review on the potential risks. International journal of environmental research and public health, 19(16), 9960.
Valkenburg, P. M., Meier, A., & Beyens, I. (2022). Social media use and its impact on adolescent mental health: An umbrella review of the evidence. Current opinion in psychology, 44, 58-68.
- What is the relevance of the study? Authors should indicate the specific contribution of the study findings to theory and practice.
We thank the referee for this question, which allowed us to clarify in the discussions the theoretical and practical contributions of the study. In particular at the practical level, we added more clearly that “the present research could be useful to identify potential adolescents as preferred recipients of peer trust for implementing peer education interventions. For example these trusty adolescents could become active figure to promote citizenship and responsible social media usage educative program. While at theoretical level the present study could offer a detailed picture of how trust in online relationships between adolescents is configured. It sheds light on the factors that contribute to the attribution of trust in virtual environments, enhancing the comprehension of this complex phenomenon
Reviewer 3 Report
Dear Authors:
the article: Adolescents and Trust in Online Social Interactions: A Qualitative
Exploratory Study.
The paper aims to explore the components of online trust of teenagers.
- qualitative methodology
Important/interesting ideas:
Social media allow adolescents to experiment with their own identities.
PROBLEM:
NUMBER THE INDIVIDUAL PARTS OF THE ARTICLE
for example:
1. Introduction
2. Materials -
2.1. The socio-cognitive model of trust
2.2. Purpose of the study
....
4. Discussion
For better understanding add these 2 articles (in discussion)
Vrabec, N., Kačinová, V., Kitsa, M., & Majda, M. (2023). Non-Formal Education Focused on the Development of Critical Thinking and Media Literacy: The Role and Activities of Key Stakeholders in Slovakia. Journal of Education Culture and Society, 14(1), 493–502. https://doi.org/10.15503/jecs2023.1.493.502
Tkáčová, H. . (2021). Forms of prejudice about christians and social cohesion between university students in Slovakia: media as an essential part of the issue. Journal of Education Culture and Society, 12(1), 429–444. https://doi.org/10.15503/jecs2021.1.429.444
The authors understand the problem raised in the article, about which they wrote a very interesting article.
Authors set the questions and answer them properly.
Very good: Conclusion
I recommend publishing the article.
Author Response
Please find below our answers to third reviewer. The changes in the manuscript were made in red.
PROBLEM:
NUMBER THE INDIVIDUAL PARTS OF THE ARTICLE
for example:
Response: we have added this limitation in the discussion.
Discussion
For better understanding add these 2 articles (in discussion)
Vrabec, N., Kačinová, V., Kitsa, M., & Majda, M. (2023). Non-Formal Education Focused on the Development of Critical Thinking and Media Literacy: The Role and Activities of Key Stakeholders in Slovakia. Journal of Education Culture and Society, 14(1), 493–502. https://doi.org/10.15503/jecs2023.1.493.502
Tkáčová, H. . (2021). Forms of prejudice about christians and social cohesion between university students in Slovakia: media as an essential part of the issue. Journal of Education Culture and Society, 12(1), 429–444. https://doi.org/10.15503/jecs2021.1.429.444
Response: We have added the article consistently with the current version of the manuscript.
The authors understand the problem raised in the article, about which they wrote a very interesting article.
Authors set the questions and answer them properly.
Very good: Conclusion
I recommend publishing the article.
We greatly appreciate the positive feedback from the reviewer

Round 2
Reviewer 1 Report
Dear Author:
Review
Adolescents and Trust in Online Social Interactions: A Qualitative
Exploratory Study
Abstract: very well processed and content balanced.
Well-structured article
The Authors analyzed the textual material deriving from the transcription of the interviews using deductive-inductive content analysis, following the procedure described.
Add 1-2 paragraphs to Reseult (p.14)
This paper seeks to expand the current literature on adolescents' relationships on the social media and trust.
Methods, references - no problem.I recommend publishing this paper.
Author Response
Dear reviewer,
Thank you for your suggestion. We have revised the article according to your request. Thanks again for your review.
Reviewer 2 Report
1. Authors claim that "although virtual contexts have been widely studied, the crucial role of trust and its potential impact on online interpersonal relationships is still not clear ".
First, authors should provide an analytical and critical analysis of virtual contexts that have been widely studied.
Second, authors need to state clearly what is not clear about the crucial role of trust and its potential impact on online interpersonal relationships and how this is not clear in literature.
2. Authors assert that "although, the literature on social media and adolescents is extensive, in the light of our current knowledge, there appears to be a lack of research on the specific elements that lead adolescents to trust others online ".
Authors should showcase the extensiveness of literature on social media and adolescents. Second, they should present the current knowledge about the phenomenon and three, stating that "there appears to be a lack of research on the specific elements that lead adolescents to trust others online" is vague. What particular elements are the researchers interested in?
3. In essence, the authors have not provided clarity on their research gap/research problem.
4. The discussion section should be separated from the conclusion section.
5. How was the instrument used in the study designed and what informed the design?
6. Where did the elements captured in the study emanate from? Literature, grounded study, or from a theoretical framework?
7. In the second line of the first paragraph in the Analysis section, change "Elo & Kyng" to "Elo and Kyng". Authors should further proofread the entire manuscript.
8. The real contribution of the study findings to theory and practice seems lacking.
Some minor editing is required.
Author Response
Thank you very much for the useful comments and suggestions, that allowed us to clarify and explicit some less clear points.
We have revised the manuscript as proposed, and hope the article has been improved in the suggested direction.
You will find responses to your comments below.
Comments and Suggestions for Authors
REV: Authors claim that "although virtual contexts have been widely studied, the crucial role of trust and its potential impact on online interpersonal relationships is still not clear ".
First, authors should provide an analytical and critical analysis of virtual contexts that have been widely studied.
Authors: We have revised this point specifying that in our study we are not referring to virtual contexts in general, but to the specific context of social media.
REV: Second, authors need to state clearly what is not clear about the crucial role of trust and its potential impact on online interpersonal relationships and how this is not clear in literature.
Authors: We have rewritten this part specifying that what is not clear in the literature is how trust is attributed in online environments, with a particular focus on social media and adolescence. The literature review made in the paragraph "Online interpersonal trust and adolescents" (which starts from the analysis of studies on online trust in adolescence and then focuses on the specific context of social media) should clarify the absence of studies in this sense.
REV: Authors assert that "although, the literature on social media and adolescents is extensive, in the light of our current knowledge, there appears to be a lack of research on the specific elements that lead adolescents to trust others online ". Authors should showcase the extensiveness of literature on social media and adolescents. Second, they should present the current knowledge about the phenomenon and three, stating that "there appears to be a lack of research on the specific elements that lead adolescents to trust others online" is vague. What particular elements are the researchers interested in?
Authors: We have added a section on the literature on social media and adolescents, which precedes the section on online trust in adolescence, where the current knowledge about the phenomenon is presented. We have specified that the elements we are interested in are social and cognitive.
REV In essence, the authors have not provided clarity on their research gap/research problem.
Authors: We hope that with the revisions we have made the gap is clearer.
REV: The discussion section should be separated from the conclusion section.
Authors: We have separated the two sections.
- How was the instrument used in the study designed and what informed the design?
This information is already in the text in the paragraph method and the sub-paragraph interview, as reported below.
From a theoretical point of view, the socio-cognitive model of trust informed the entire study, starting from the research questions and from the definition of the interview outline, followed by the data analysis up to the reading of the results (method paragraph).
In particular, researchers developed an ad-hoc interview guide following the interview steps by Smith and collaborators (Smith, Langenhove, & Harre, 1995), and paying particular attention to the formulation of the questions, as suggested by McNamara (2009). The definition of the questions had as a theoretical reference the socio-cognitive model of trust and was aimed at exploring the model's beliefs and how they are applied in online relationships between adolescents (interview paragraph).
REV Where did the elements captured in the study emanate from? Literature, grounded study, or from a theoretical framework?
Authors: We have inserted the following sentence in the paragraph on data analysis to better specify this aspect: “Therefore the elements captured in the study are partly emanated from the referring theoretical framework and partly from collected data”.
REV. In the second line of the first paragraph in the Analysis section, change "Elo & Kyng" to "Elo and Kyng". Authors should further proofread the entire manuscript.
Authors: We have replaced the & with and in the entire manuscript.
REV. The real contribution of the study findings to theory and practice seems lacking.
Authors: We have tried to clarify this aspect better.
REV: Comments on the Quality of English Language
Some minor editing is required.
Authors: We further edited the manuscript, correcting mistakes that had actually escaped us.
Round 3
Reviewer 2 Report
1. Is it that there is completely a lack of research on the social and cognitive elements that lead adolescents to trust others online or that such research is still emerging and at infancy stage, implying that a few studies have been done on the phenomenon? This is not clearly stated in the study background.
2. The authors allude that "even with the increasing dissemination of studies on online trust, how trust is attributed in online environments, with a particular focus on social media and adolescence, is still unknown or not sufficiently investigated". Authors should provide a rationale for the assertion with accompanying citations.
3. Authors should indicate the conceptual framework for the study and clearly state how that relates to the theoretical framework and the entire study.
4. What is the practical implications and contributions of the study findings?
Minor English language editing required.
Author Response
- Is it that there is completely a lack of research on the social and cognitive elements that lead adolescents to trust others online or that such research is still emerging and at infancy stage, implying that a few studies have been done on the phenomenon? This is not clearly stated in the study background.
Authors. We have better specified the completely lack of research, as you can se in the Introduction: “Although the literature on social media and adolescence is extensive (Bozzola et al. 2022; Valkenburg et al. 2021), in the light of our current knowledge, there is a completely lack of research on the social and cognitive elements that lead adolescents to trust others online” and in the “Online interpersonal trust and adolescents” paragraph: “As shown in this literature review, although the increasing dissemination of studies on online trust, how trust is attributed in online environments, with a particular focus on social media and adolescence, is still unknown”.
- The authors allude that "even with the increasing dissemination of studies on online trust, how trust is attributed in online environments, with a particular focus on social media and adolescence, is still unknown or not sufficiently investigated". Authors should provide a rationale for the assertion with accompanying citations.
Authors. We have revised the sentence in this way: “As shown in this literature review, although the increasing dissemination of studies on online trust, how trust is attributed in online environments, with a particular focus on social media and adolescence, is still unknown”.
The paragraph “Online interpersonal trust and adolescents” already contains an analysis of the literature on the subject, with all research identified. We hope that with the sentence revised this aspect is now more clear.
- Authors should indicate the conceptual framework for the study and clearly state how that relates to the theoretical framework and the entire study.
We had already replied to your observations of a methodological nature, integrating them into the article text. We are surprised that further methodological requests are coming in the third round. We believe we have described everything that can be explained with respect to a qualitative study in the "Methodology" section, and the respective sub-paragraphs. We apologize but we really don't know what else to add.
- What is the practical implications and contributions of the study findings?
Authors. The practical implications have already been included in the conclusion paragraph “From an applicative point of view, knowing these factors can be useful for supporting young people's relationships on social media. Specifically, the results of the present research could be used for developing awareness-raising interventions on the risks that adolescents are exposed to when they trust others in an online context, in order to promote “safe” relationships of trust. For example, starting from factors that led adolescents to trust others online, it is possible to identify potential adolescents as preferred recipients of peer trust for implementing peer education interventions. These trusty adolescents could become active figure to promote citizenship and responsible social media usage educative program”. The theoretical contribution of the study has been inserted too: “At theoretical level the present study offers a detailed picture of how trust in online relationships between adolescents is configured. It sheds light on the social and cognitive factors that contribute to the attribution of trust in virtual environments, enhancing the comprehension of this complex phenomenon. On one hand, the study contributed to the advancement of the theoretical reference model, showing its potential application in online trust attribution processes. From the other hand, the study contributed to fill the literature gap not only concerning the knowledge of factors on which adolescents base their trust on social media, but also concerning what these factors mean for young people”.